# Margelopsid species search taxonomic home within Corymorphidae and Boreohydridae

Daria Kupaeva[1], Tatiana Lebedeva[2], Zachariah Kobrinsky[3], Daniel Vanwalleghem[4], Andrey Prudkovsky[5] and Stanislav Kremnyov[1,6]

[1] Department of Embryology, Faculty of Biology, Lomonosov Moscow State University, Moscow, Russia
[2] Department of Neurosciences and Developmental Biology, Faculty of Life Sciences, University of Vienna, Vienna, Austria
[3] Unaffiliated, Independent Wildlife Photographer, New York City, United States of America
[4] Plankton Monitoring Station, Ostend, Belgium
[5] Department of Invertebrate Zoology, Faculty of Biology, Lomonosov Moscow State University, Moscow, Russia
[6] Laboratory of Morphogenesis Evolution, Koltzov Institute of Developmental Biology RAS, Moscow, Russia

## ABSTRACT

Planktonic lifestyle of polyps in representatives of Margelopsidae are very different from all other species in the hydrozoan clade Aplanulata. Their evolutionary origin and phylogenetic position have been the subject of significant speculation. A recent molecular study based only on COI data placed Margelopsidae as a sister group to all Aplanulata, an unexpected result because margelopsid morphology suggests affiliation with Tubulariidae or Corymorphidae. Here we used multigene analyses, including nuclear (18S rRNA and 28S rRNA) and mitochondrial (16S rRNA and COI) markers of the hydroid stage of the margelopsid species *Margelopsis haeckelii* and the medusa stage of *Margelopsis hartlaubii* to resolve their phylogenetic position with respect to other hydrozoans. Our data provide strong evidence that *M. haeckelii*, the type species of *Margelopsis*, is a member of the family Corymorphidae. In contrast, *M. hartlaubii* is sister to *Plotocnide borealis*, a member of Boreohydridae. These results call into question the validity of the genus *Margelopsis* and the family Margelopsidae. The systematic position of *M. haeckelii* is discussed in light of the phylogeny of Corymorphidae.

## INTRODUCTION

Species in the family Margelopsidae Mayer, 1910 (Aplanulata, Hydrozoa, Cnidaria) have intriguing life histories. The family is exclusively represented by hydrozoans with holopelagic life-cycles, where medusae and solitary vasiform polyps float freely throughout the water column. Although dispersion by polyps is widespread among hydrozoans Hydroidolina, it usually happens by accident (*Cabral et al., 2015*). The most specialized planktonic hydroids related to Porpitidae and Siphonophorae have morphologically complex, polymorphic colonies that do not come into contact with the bottom during their life cycle (*Kirkpatrick & Pugh, 1984*). Interestingly, siphonophore specialists used

Corresponding authors
Andrey Prudkovsky,
aprudkovsky@wsbs-msu.ru
Stanislav Kremnyov,
s.kremnyov@gmail.com



PeerJ Hubs
Published on behalf of

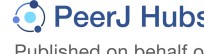
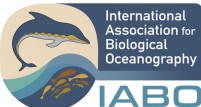

margelopsid species as a model to explain the origin of siphonophoran colonies (*Totton & Bargmann, 1965*).

Margelopsidae is comprised of three genera: *Margelopsis* (*Hartlaub, 1897*); *Pelagohydra* Dendy, 1902; and *Climacocodon* Uchida, 1924, none of which have been sampled for comprehensive molecular analyses. Phylogenetic analysis using only COI sequences (*Ortman et al., 2010*) of *Margelopsis hartlaubii* (*Browne, 1903*) suggested that Margelopsidae might be the sister group to the rest of Aplanulata (*Nawrocki et al., 2013*). However, authors have not received strong support for this placement. Moreover, the authors attribute planktonic polyps to *M. hartlaubii*, although the life cycle of this species has not been described (*Nawrocki et al., 2013*). Given their polyp morphology, species of Margelopsidae show affinities with Tubulariidae or Corymorphidae, but the margelopsid medusa is unique for Tubularioidea, having several tentacles grouped together on each bulb. It was used to justify their original erection as a separate family. Thus, sampling with more DNA markers and specimens—especially including the type species *Margelopsis haeckelii* (*Hartlaub, 1897*)—has been needed to determine the scope and phylogenetic position of the family Margelopsidae.

Despite difficulties of sampling margelopsid hydroids, we were finally able to collect representatives of *Margelopsis haeckelii* and *Margelopsis hartlaubii* for molecular studies. *Margelopsis haeckelii* is the most studied species of its family, yet, documented collection records and morphological examinations have been very few (*Hartlaub, 1897*; *Hartlaub, 1899*; *Leloup, 1929*; *Werner, 1955*; *Schuchert, 2006*). Polyps of *M. haeckelii* closely resemble tubulariid hydranths, having two whorls of tentacles but lacking both a hydrocaulus and stolonal system (Figs. 1A and 1B). Free-swimming medusae develop from medusa buds located between whorls of polyp tentacles (Figs. 1B, 1C and 1D). Eggs of *M. haeckelii* develop on the manubrium of the medusa (Figs. 1C and 1D) and transform directly or through an encysted stage into a hydranth that never fixes to a substrate, exhibiting a continuous planktonic lifestyle (*Werner, 1955*). It is thought that eggs of this species are parthenogenetic, as no male gonads have ever been reliably documented (*Werner, 1956*; *Schuchert, 2006*). The only hermaphrodite specimen of *M. haeckelii* is known without detailed description (*Werner, 1956*). There is less information about *M. hartlaubii*, which is only known from the medusa stage. The medusa of *M. hartlaubii* can readily be distinguished from the medusa of *M. haeckelii* by its thick apical mesoglea of the bell without apical canal and 2–3 tentacles per bulb (Figs. 1C, 1D and 1E) (*Schuchert, 2006*).

In our study we obtained full-length sequences of 18S rRNA and 28S rRNA and partial sequences of the mitochondrial ribosomal 16S rRNA and cytochrome oxidase subunit I (COI) in order to phylogenetically place *M. haeckelii* and *M. hartlaubii* within as comprehensive sampling of Aplanulata hydrozoan taxa as possible. Using this approach, we provide the first molecular evidence that *M. haeckelii* should be placed within the family Corymorphidae. Our findings further showed that the previously sequenced *M. hartlaubii* is a relative of the family Boreohydridae, and is only distantly related to *Margelopsis haeckelii*, the type species of the genus. Portions of this text were previously published as part of a preprint (*Kupaeva et al., 2022*).

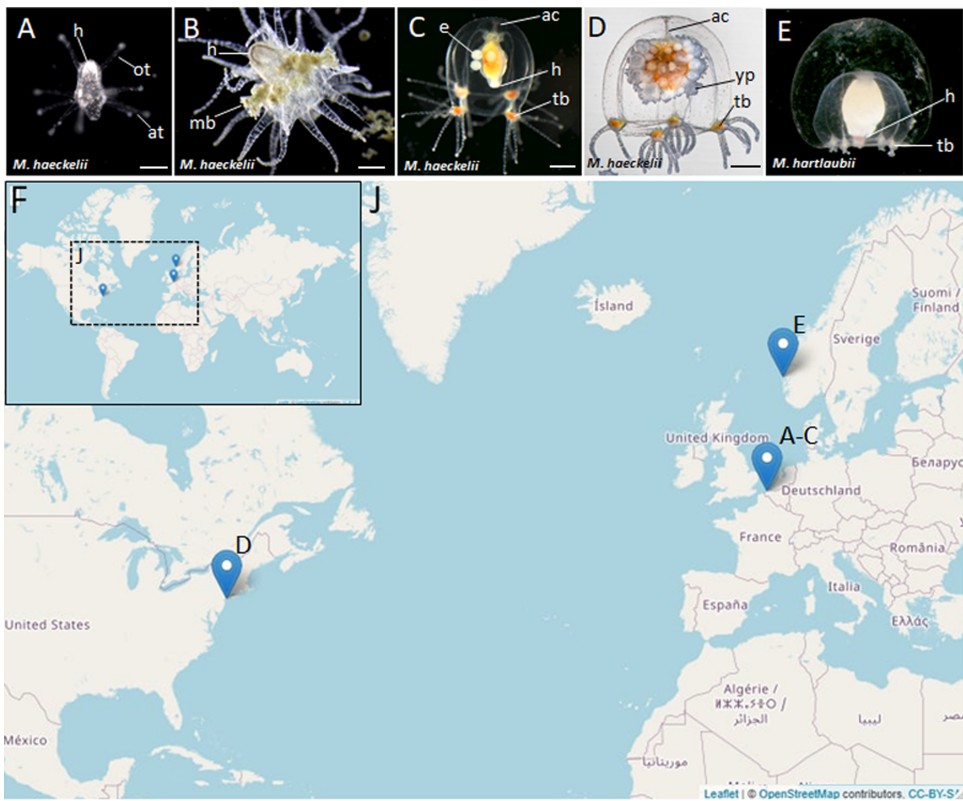

**Figure 1** **Morphology of collected Margelopsidae representatives and the locations of its samplings.**
(A–D) *Margelopsis haeckelii* (*Hartlaub, 1897*). (A) Newly hatched polyp, (B) Mature polyp with medusa buds, (C, D) Mature medusa. (E) Mature medusa of *Margelopsis hartlaubii* (*Browne, 1903*). Photo Credit: Dr. Peter Schuchert. (F, J) Geographic locations of sampling sites. Abbreviations: ac, apical canal; at, aboral tentacles; e, embryos; h, hypostome; mb, medusa bud; ot, oral tentacles; tb, tentacle bulb; yp, young polyp. Scale bars: A (200 µm), B–D (500 µm).

## METHODS AND MATERIALS

### Animal sampling

Nine *M. haeckelii* polyps were collected in the North Sea (loc. Belgium, Ostend, 51.218028°, 2.879417°) (Figs. 1F and 1J). Polyps were collected with an Apstein plankton net (diameter of the net ring is 30 cm, mesh size 100 µm) in the coastal area. Collected animals were used to set up a lab culture. The obtained culture was maintained throughout the year in aquaria using artificial sea water (salt Red Sea Coral Pro, salinity 30–32‰) at the Department of Embryology, Lomonosov Moscow State University, Moscow, Russia. For both polyp and medusa stages, *Artemia salina* nauplii, at least 3 days after hatching, were used for feeding. Animals were fed once a day.

Also, two *M. haeckelii* medusae were collected in the Atlantic Ocean, Atlantic Coast of North America (loc. USA, New York, 40.560556°, −73.882333°). Medusae were collected with an Apstein plankton net (diameter of the net ring is 35 cm, mesh size 50 µm) in the coastal area, about 10 m out from the shore. Collected animals were fixed and stored in

96% ethanol (Figs. 1F and 1J). Sampled animals have been identified as *Margelopsis* sp. based on morphological characters (*Schuchert, 2006*).

*M. hartlaubii* DNA has been provided by Dr. Peter Schuchert. *M. hartlaubii* medusae specimens were collected in the North Sea (loc. Norway, Raunefjord, 60.2575°, 05.1393°) with a plankton net from 200 to 0 m depth. Sampled medusae were fixed and stored in 96% ethanol (Figs. 1F and 1J).

Meiobenthic polyps of *Plotocnide borealis* (formerly known as *Boreohydra simplex*; *Pyataeva et al., 2016*) were collected in the White Sea near the N.A. Pertsov White Sea Biological Station of the Moscow State University, Kandalaksha Bay, Russia (66.528056°, 33.185556°). Fine mud with polyps was collected with a light hyperbenthic dredge from depth 20-40 m. Collected individuals were fixed and stored in 96% ethanol. The polyps were collected at the same location as in *Pyataeva et al. (2016)* and their external morphology fully corresponded to the description.

## Identification of COI, 16S rRNA, 18S rRNA and 28S rRNA sequences

COI, 16S rRNA, 18S rRNA and 28S rRNA sequence fragments were amplified from genomic DNA using PCR methods. Genomic DNA was extracted from single animals using standard phenol/chloroform protocols. This method involved tissue digestion with proteinase K (20 mg/mL) (cat# EO0491; Thermo Fisher Scientific, Waltham, MA, USA) in a lysis buffer (20 mM Tris-CL pH 8.0, 5 mM EDTA pH 8.0, 400 mM NaCl, 2%SDS) (ON at 37 °C), extraction with phenol/chloroform (1:1), precipitation with 0.1 vol 3M sodium acetate and 1 vol. 100% isopropanol. Precipitated gDNA has been centrifuged for 20 min at 14,500 rpm. Supernatant has been decanted, obtained pellet has been air-dried and dissolved in mQ water.

For amplification, we used the following primers pairs:

16SAR (TCGACTGTTTACCAAAAACATAGC) and 16SBR (ACGGAATGAACT-CAAATCATGTAAG) for 16S rRNA (*Cunningham & Buss, 1993*); and jGLCO1490 (TITCIACIAAYCAYAARGAYATTGG) and jGHCO2198 (TAIACYTCIGGRTGIC-CRAARAAYCA) for COI (*Geller et al., 2013*). Amplification programs used for 16S rRNA and COI are as previously described in *Prudkovsky, Ekimova & Neretina (2019)*.

18S-EukF (WAYCTGGTTGATCCTGCCAGT) and 18S-EukR (TGATCCTTCYGCAGG TTCACCTAC) for 18S rRNA (*Medlin et al., 1988*). F97(CCYYAGTAACGGCGAGT), R2084 (AGAGCCAATCCTTTTCC), F1383(GGACGGTGGCCATGGAAGT) and R3238 (SWACAGATGGTAGCTTCG) for 28S rRNA (*Evans et al., 2008*). Amplification programs used for 18S rRNA and 18S rRNA are as previously described in *Evans et al. (2008)*.

Full-length 18S rRNA and 28S rRNA sequences of *M. haeckelii* from the North Sea were obtained from the reference transcriptome available in our laboratory. For transcriptome sequencing, total RNA was extracted from a mixture of various *Margelopsis* life and developmental stages. Total RNA extraction was conducted using the Zymo Research Quick-RNA MiniPrep Plus Kit according to the manufacturer's instructions. Poly-A RNA enrichment, cDNA library construction and sequencing were carried out at Evrogen (Russia). The cDNA library was sequenced using the Illumina NovaSeq 6000 SP flow cell to produce 150-bp paired-end reads. The high-quality reads were employed for the *M.*

*haeckelii* transcriptome assembly with the SPAdes assembler (v.3.13.1) (*Bankevich et al., 2012*).

## Phylogenetic analyses

Nucleotide sequences were aligned using the MUSCLE algorithm (*Edgar, 2004*) in MEGA 6 software (*Tamura et al., 2013*).

Phylogenetic analyses were performed using Maximum Likelihood methods in IQTree v.2.0-rc2 software (*Minh et al., 2020*) according to the optimal models for each gene. Outgroups were selected according to *Nawrocki et al. (2013)*. Individual marker analyses and a concatenated gene analysis were performed. The best models of nucleotide substitution were chosen using ModelFinder (*Kalyaanamoorthy et al., 2017*). The GTR+F+I+G4 was found to be optimal for the COI dataset according to Bayesian information criterion (BIC); GTR+F+I+G4 for 16S rRNA; TIM3+F+R3 for 18S rRNA; and TIM3+F+R4 for 28S rRNA. One thousand bootstrap replicates were generated for each individual analysis, as well as for the combined analysis.

The concatenated COI+16S+18S+28S alignment was constructed using Sequence Matrix (https://github.com/gaurav/taxondna). The concatenated dataset was analyzed using IQTree (v.2.0-rc2) with partitioned analysis for multi-gene alignments (*Chernomor, Von Haeseler & Minh, 2016*).

Bayesian phylogenetic trees were built as described by us previously in *Prudkovsky et al. (2023)* and *Prudkovsky, Vetrova & Kremnyov (2023)*. In brief: Bayesian phylogenetic trees were built in PhyloBayes 3.3 (*Lartillot, Lepage & Blanquart, 2009*) according to the optimal models for each gene. JModelTest 2 (*Darriba et al., 2012*) with minimum number of substitution schemes was used to estimate the best substitution model for each partition based on the Bayesian information criterion (BIC). The GTR+I+G was found to be optimal for the COI, 18S and 28S datasets; HKI+I+G for 16S rRNA. Two MCMC chains were run in parallel, and the analyses were stopped when the maximum discrepancy of bipartitions between chains was below 0.01. We used a Tracer 1.7.2 (*Rambaut et al., 2018*) to analyze convergence of MCMC chains.

Trees were visualized in FigTree v1.4.4. Font and color of the labels were edited in Adobe Illustrator CC 2015.

Uncorrected *p*-distances were calculated with MEGA 6 software with option «pairwise deletion» (*Tamura et al., 2013*).

Parsimony ancestral character state reconstruction of medusa symmetry was conducted in Mesquite v3.81 (*Maddison & Maddison, 2023*). Bayes Interference consensus tree for multigene dataset was used and three character states (absent free-swimming medusa, radial medusa and bilateral medusa) were assigned based on published literature.

## Data availability

Sequences obtained in this study have been deposited in GenBank under the following accession numbers: *Margelopsis haeckelii* (OK129327, OK139084, OK142735, OK127861, ON391039, ON391070), *Margelopsis hartlaubii* (ON237369, ON237671, ON237710), *Plotocnide borealis* (OK110252).
Portions of this text were previously published as part of a preprint (*Kupaeva et al., 2022*).

## RESULTS

Our investigation of phylogenetic affinities of species of Margelopsidae was conducted employing both Maximum Likelihood and Bayesian analysis for all single gene datasets as well as our final concatenated four-gene dataset (COI, 16S rRNA, 18S rRNA, 28S rRNA). All taxa used in our analysis are arranged taxonomically in Table 1. All *Margelopsis haeckelii* and *M. hartlaubii* sequences (COI, 16S rRNA, 18S rRNA, 28S rRNA) were newly generated for this study. *M. hartlaubii* had previously only had COI available on GenBank (GQ120058.1) (*Ortman et al., 2010*). Consistent results were obtained for different datasets. In single-gene trees the specimens of *M. haeckelii* form a clade with *Corymorpha* spp., however, this clade is not supported except for 28S ($p = 1$; ML $= 100$). Both the Bayesian inference and Maximum Likelihood analysis of the concatenated dataset recovered a relatively well-resolved tree and recovered Margelopsidae paraphyly. *M. hartlaubii* was recovered as sister to *Plotocnide borealis* ($p = 1$; MLB $= 100$), forming a clade that affiliates with the family Boreohydridae, a sister taxon to all other Aplanulata genera used in our phylogenetic analysis ($p = 1$; MLB $= 100$) (Fig. 2). Each individual COI, 16S rRNA, 18S rRNA or 28S rRNA analysis also recovered a strong supported affiliation of *M. hartlaubii* within Boreohydridae ($p = 1$; MLB $= 100$) (Figs. 1S, 2S, 4S and 4S). At the same time, both *M. haeckelii* from different locations nested within the main clade of the Corymorphidae ($p = 1$; MLB $= 80$). Corymorphidae was found to be a polyphyletic group with *Corymorpha groenlandica* (Allman, 1876) and *Hataia parva* Hirai & Yamada, 1965 nested out of the main corymorphid clade. The main clade comprised two subclades, each well supported, one for the genus *Euphysa*, including the type species *Euphysa aurata* Forbes, 1848, and the other for *Corymorpha + M. haeckelii*, including the type species, *Corymorpha nutans* (*Sars, 1835*) (Fig. 2). *Margelopsis haeckelii* is nested within the clade *Corymorpha bigelowi* Maas, 1905, *Corymorpha nutans*, *Corymorpha sarsii* Steenstrup, 1855, *Corymorpha glacialis* Sars, 1860 and *Corymorpha pendula* L. Agassiz, 1862. Clade *Euphysa+Corymorpha+M. haeckelii* was recovered to be the sister to Tubulariidae. Tubulariidae and Corymorphidae (with the exclusion of *C. groenlandica* and *Hataia parva*) together with *Branchiocerianthus imperator* Allman, 1885 constitute the superfamily Tubularioidea. Tubularioidea is recovered as sister to Hydridae ($p = 1$; ML $= 86$). General topology of our phylogenetic tree obtained in combined analysis coincides with the Aplanulata tree published by *Nawrocki et al. (2013)*.

Some *Corymorpha* species that were previously absent in the molecular phylogenetic analyses were included in our datasets: *Corymorpha gracilis* (Brooks, 1883), *Corymorpha floridana* Schuchert & Collins, 2021, and *Corymorpha forbesii* (Mayer, 1894) in 16S-dataset (Fig. 2S), and *Corymorpha verrucosa* (Bouillon, 1978) in COI-dataset (Fig. 1S). *Corymorpha verrucosa* apparently groups with the clade *Euphysa* although without significant support (Fig. 1S: $p = 0.6$; MLB $= 56$). *Corymorpha gracilis* and *C. floridana* forms a clade (Fig. 2S: $p = 1$; MLB $= 100$) which is nested within the main corymorphid clade (Fig. 2S: $p = 0.79$; MLB $= 27$). *Corymorpha forbesii* and *C. bigelowi* form a clade (Fig. 2S: $p = 1$; MLB $= 100$) unrelated with the main corymorphid clade.
**Table 1  List of the species included in the study and corresponding GenBank accession numbers of all analyzed sequences.**

| Suborder | Family | Species | 16S rRNA | 18S rRNA | 28S rRNA | COI | Vouchers |
|---|---|---|---|---|---|---|---|
| Aplanulata | Boreohydridae | *Plotocnide borealis* | KU721822.1 | KU721833.1 | OK110252 | KU721812.1 | RU087.2 |
| | Candelabridae | *Candelabrum cocksii* | EU876535.1 | AY920758.1 | AY920796.1 | GU812438.1 | MHNGINVE29591 |
| | | *Candelabrum beringensis* | OP895642 | OP895665 | OP895649 | | MIMB 44079 |
| | | *Candelabrum sp* | EU876530 | EU876557 | EU879929 | JX121579 | PC0003 |
| | Corymorphidae | *Branchiocerianthus imperator* | | JN594046.2 | JN594035.2 | JX121580.1 | MHNG:INVE 74105 |
| | | *Branchiocerianthus radialis* | OP895644 | OP895664 | OP895650 | | MIMB 44081 |
| | | *Corymorpha bigelowi* | EU448099 | EU876564.1 | EU272563.1 | JX121581.1 | KUNHM 2829 |
| | | *Corymorpha anthoformis* | JX122502 | | JX122504 | | |
| | | *Corymorpha floridana* | MW528714 | | | | |
| | | *Corymorpha forbesii* | MW528642 | | | | |
| | | *Corymorpha glacialis* | FN687549 | JN594047 | JN594036 | JX121584 | |
| | | *Corymorpha gracilis* | MW528715 | | | | |
| | | *Corymorpha groenlandica* | FN687551 | JN594048 | JN594037 | | |
| | | *Corymorpha nutans* | EU876532.1 | EU876558.1 | EU879931.1 | JX121586.1 | MHNG:INVE 48745 |
| | | *Corymorpha pendula* | EU876538.1 | EU876565.1 | EU305510.1 | JX121583.1 | KUNHM DIZ2962 |
| | | *Corymorpha sarsii* | KP776787.1 | JN594049.2 | JN594038.2 | JX121585.1 | MHNG:INVE 68950 |
| | | *Corymorpha verrucosa* | | | | JQ716061 | |
| | | *Euphysa aurata* | EU876536.1 | EU876562.1 | EU879934.1 | JX121587.1 | MHNG:INVE 48753 |
| | | *Euphysa intermedia* | EU876531.1 | AY920759.1 | EU879930.1 | JX121582.1 | |
| | | *Euphysa japonica* | KP776802.1 | EU301605.1 | JX122505.1 | MF000498.1 | |
| | | *Euphysa tentaculata* | EU876537.1 | EU876563.1 | EU879935.1 | JX121588.1 | |
| | | *Hataia parva* | JN594033.1 | JN594045.2 | JN594034.2 | JX121608.1 | UF:5407 |
| | Hydridae | *Hydra canadensis* | JF884206.1 | | | JN594039.2 | |
| | | *Hydra circumcincta* | | EU876568.1 | AY026371.1 | MF135312.1 | |
| | | *Hydra hymanae* | GU722762.1 | JN594051.2 | JN594040.2 | GU722849.1 | |
| | | *Hydra oligactis* | | JN594052.2 | JN594041.2 | GU722871.1 | |
| | | *Hydra utahensis* | | JN594053.2 | JN594042.2 | GU722861.1 | |
| | | *Hydra vulgaris* | EU876543.1 | JN594054.2 | JN594043.2 | GU722914.1 | |
| | | *Hydra viridissima* | | EU876569.1 | EU879940.1 | GU722845.1 | |

**Table 1** (*continued*)

| Suborder | Family | Species | 16S rRNA | 18S rRNA | 28S rRNA | COI | Vouchers |
|---|---|---|---|---|---|---|---|
| | Margelopsidae | *Margelopsis haeckelii* | OK129327 ON391070 | OK139084 | OK142735 | OK127861 ON391039 | |
| | | *Margelopsis hartlaubii* | ON287278 | ON237671 | ON237710 | ON391039 GQ120058.1 | |
| | Protohydridae | *Protohydra leuckarti* | KU721828.1 | KU721835.1 | | KU721813.1 | Protohydra20100727.6 |
| | Tubulariidae | *Ectopleura crocea* | EU876533.1 | KF699111.1 | EU879932.1 | JX121589.1 | MHNG:INVE 34010 |
| | | *Ectopleura dumortierii* | FN687542.1 | EU876561.1 | EU879933.1 | JX121590.1 | |
| | | *Ectopleura larynx* | | EU876572.1 | EU879943.1 | JX121591.1 | MHNG-INVE-54563 |
| | | *Ectopleura marina* | EU883542.1 | EU883547.1 | EU883553.1 | JX121592.1 | |
| | | *Ectopleura wrighti* | FN687541.1 | JN594055.2 | JN594044.2 | JX121593.1 | MHNG:INVE 27331 |
| | | *Hybocodon chilensis* | EU876539.1 | EU876566.1 | EU879937.1 | JX121594.1 | MHNG:INVE 36023 |
| | | *Hybocodon prolifer* | FN687544.1 | EU876567.1 | EU879938.1 | JX121595.1 | |
| | | *Ralpharia gorgoniae* | EU305482.1 | EU272633.1 | EU272590.1 | GU812437.1 | KUNHM2778 |
| | | *Tubularia indivisa* | FN687530.1 | EU876571.1 | EU879942.1 | JX121596.1 | |
| | | *Tubularia sp* | OP895640 | OP895668 | OP895647 | | MIMB 44077 |
| | | *Zyzzyzus warreni* | EU305489.1 | EU272640.1 | EU272599.1 | JX121597.1 | KUNHM 2777 |
| Capitata | Asyncorynidae | *Asyncoryne ryniensis* | EU876552.1 | EU876578.1 | GQ424289.1 | | KUNHM 2639 |
| | Cladocorynidae | *Cladocoryne floccosa* | AY512535.1 | EU272608.1 | EU272551.1 | | personal:A. Lindner:AL1407 |
| | Cladonematidae | *Staurocladia vallentini* | GQ395332.1 | GQ424322.1 | GQ424293.1 | MF000500.1 | Sch522 |
| | | *Staurocladia wellingtoni* | AY787882.1 | GQ424323.1 | EU879948.1 | MF000486.1 | |
| | Corynidae | *Coryne uchidai* | GQ395319.1 | GQ424332.1 | GQ424305.1 | KT981912.1 | |
| | | *Sarsia tubulosa* | EU876548.1 | EU876574.1 | EU879946.1 | | MHNGINV35763 |
| | | *Stauridiosarsia ophiogaster* | EU305473.1 | EU272615.1 | EU272560.1 | | KUNHM 2803 |
| | Moerisiidae | *Odessia maeotica* | GQ395324.1 | GQ424341.1 | GQ424314.1 | | MHNG INVE53642 |
| | Pennariidae | *Pennaria disticha* | AM088481.1 | GQ424342.1 | GQ424316.1 | | MHNG INVE29809 |

**Table 1** (*continued*)

| Suborder | Family | Species | 16S rRNA | 18S rRNA | 28S rRNA | COI | Vouchers |
|---|---|---|---|---|---|---|---|
| | Porpitidae | *Porpita porpita* | AY935322.1 | GQ424319.1 | EU883551.1 | LT795124.1 | RM3_747 |
| | Solanderiidae | *Solanderia secunda* | EU305484.1 | AJ133506.1 | EU305533.1 | JX121599.1 | KUNHM 2611 |
| | Zancleidae | *Zanclea costata* | EU876553.1 | EU876579.1 | EU879951.1 | | MHNGINV26507 |
| | | *Zanclea prolifera* | EU305488.1 | EU272639.1 | EU272598.1 | | KUNHM 2793 |
| Fillifera | Eudendriidae | *Eudendrium capillare* | AY787884.1 | | EU305514.1 | JX121602.1 | KUNHM2625 |
| | Hydractiniidae | *Hydractinia sp* | EU305477.1 | EU305495.1 | EU305518.1 | | KUNHM2876 |
| | Proboscidactylidae | *Proboscidactyla flavicirrata* | EU305480.1 | EU305500.1 | EU305527.1 | JX121600.1 | USNM:1074994 |
| | Ptilocodiidae | *Hydrichthella epigorgia* | EU305478.1 | EU272622.1 | EU272569.1 | JX121601.1 | KUNHM 2665 |
| | Stylasteridae | *Lepidopora microstylus* | EU645329.1 | EU272644.1 | EU272572.1 | JX121603.1 | USNM:1027724 |

Minimal *P-distances* were found between *M. hartlaubii* and *P. borealis* for all genes (Table 2). Minimal *P-distances* were found between *M. haeckelii* and *C. nutans* (COI, 28S), *C. anthoformis* (16S) and *C. pendula* (18S).

Separate COI and 16S rRNA analysis recovered that individuals of *Margelopsis* from the opposite sides of the Atlantic Ocean are representatives of the same species *M. haeckelii* (Figs. 1S and 2S). No nucleotide substitutions were identified in analyzed sequences of *M. haeckelii* from the waters of Belgium(51.218028°, 2.879417°) and the USA (40.560556°, −73.882333°).

At the same time, *M. hartlaubii* COI sequences analysis revealed five mismatches between sequences obtained in this study (ON237369) and a sequence published in *Ortman et al. (2010)* (GQ120058) (Fig. 1S). However, COI sequences of *M. hartlaubii* published in *Ortman et al. (2010)* (GQ120058 and GQ120059) also are not identical and have three mismatches.

Portions of this text were previously published as part of a preprint (*Kupaeva et al., 2022*).

# DISCUSSION

## Phylogenetic position of Margelopsis haeckelii within Corymorphidae

Our concatenated dataset (COI+16S+18S+28S), which included a comprehensive taxonomic sampling of hydrozoans Aplanulata, recovered *M. haeckelii* within Corymorphidae, nested within a clade consisting of several *Corymopha* species. This result is consistent with previous findings based solely on polyp morphology, where Margelopsidae was grouped with Tubulariidae and Corymorphidae in the superfamily Tubularoidea (*Rees, 1957*). Being quite small (1–2 mm), hydrocaulus-lacking pelagic polyps of the Margelopsidae are similar to those sessile polyps of corymophids and tubulariids despite the latter having a well-developed hydrocaulus and reaching up to ten centimeters or more in height. For all three families, hydranth tentacles are arranged into two, oral and aboral

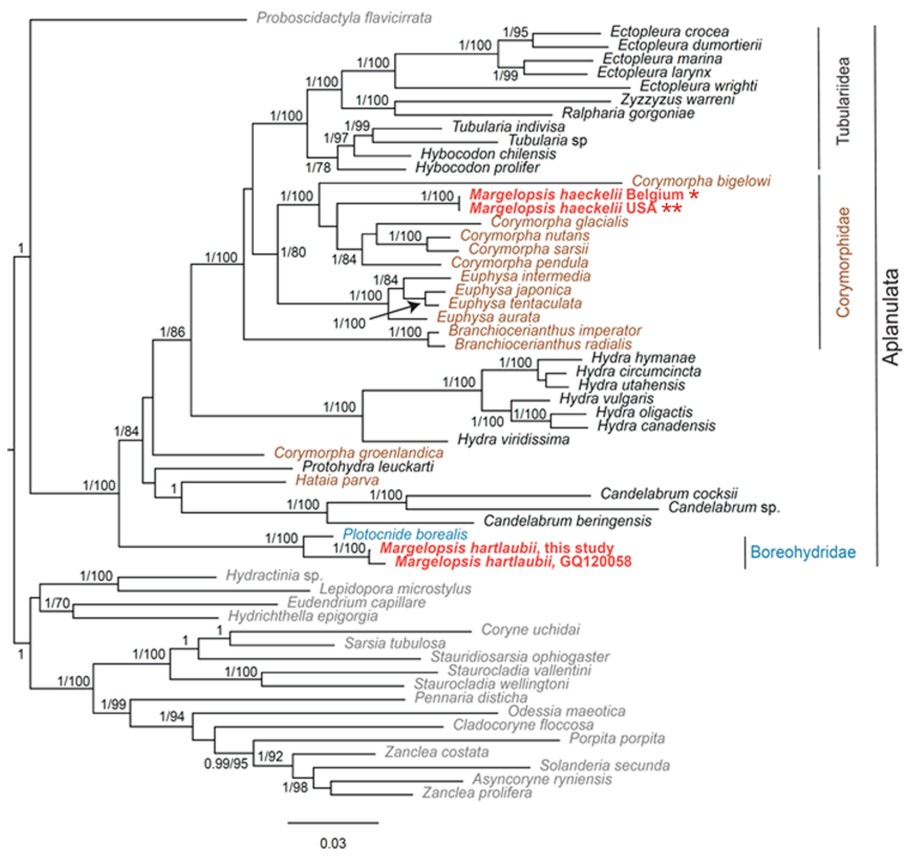

**Figure 2** **Analysis of phylogenetic position of *Margelopsis haeckelii* and *Margelopsis hartlaubii* in Aplanulata.** Bayesian and Maximum Likelihood phylogenetic hypotheses of *Margelopsis haeckelii* and *Margelopsis hartlaubii* relationships based on the combined mitochondrial and nuclear dataset (CO1 +16S +18S +28S). Node values indicate posterior probabilities ($p > 0.95$) and bootstrap values (ML >70). *Margelopsis haeckelii* and *Margelopsis hartlaubii* are in red. *WGS84 51.218028°, 2.879417°, ** WGS84 40.560556°, −73.882333°.

whorls, and blastostyles are situated in the inter-tentacular region (Figs. 3A and 3C). Our phylogenetic data support assertions that polyp tentacle patterns may be an important morphological character for identifying lineages in Aplanulata (*Rees, 1957*; *Nawrocki et al., 2013*). Hydroid *M. haeckelii* is similar to small corymorphids, which have two whorls of tentacles and lack long hydrocaulus. According to *Rees (1957)* and *Millard (1975)*, the evolution of Corymorphidae goes from simply arranged polyps with two whorls of tentacles and a poorly developed perisarc to larger and more complexly arranged forms. *Rees (1957)* considered the simplest kind of hydranth is found in *Euphysa peregrina* (Murbach, 1899) and *Gymnogonos obvolutus* (Kramp, 1933). Structural modifications accompany gigantism of large polyps such as *C. nutans* and *B. imperator* (*Rees, 1957*).

Interestingly, *M. haeckelii* jellyfish is atypical in having radial symmetry, which more usually is bilateral in Aplanulata. Radially symmetrical jellyfish are also known for the medusa of *Paraeuphysilla taiwanensis Xu, Huang & Guo, 2011*, which has four equally developed tentacles (*Wang et al., 2011*), as well as some species of *Euphysa* (Table S2:

**Table 2  *P-distances* between *Margelopsis* spp. and some species. The smallest values are indicated in bold.**

| Species | Species | *P*-distances (%) | | | |
|---|---|---|---|---|---|
| | | COI | 16S | 18S | 28S |
| *Margelopsis haeckelii* Belgium | *Corymorpha anthoformis* | – | **9.5** | – | 3.3 |
| | *Corymorpha bigelowi* | 18.1 | 20 | 2 | 5.2 |
| | *Corymorpha forbesii* | – | 18.4 | – | – |
| | *Corymorpha nutans* | **14.2** | 11.5 | 1.3 | **2.4** |
| | *Corymorpha pendula* | 15.5 | 11.3 | **1** | 2.6 |
| | *Euphysa aurata* | 15.1 | 15.2 | 1.7 | 4 |
| | *Euphysa japonica* | 17.2 | 15.2 | 1.6 | 3.9 |
| | *Margelopsis hartlaubii* | 19 | 20.2 | 3.4 | 5.6 |
| | *Plotocnide borealis* | 18.4 | 19.9 | 2.1 | 5.4 |
| *Margelopsis hartlaubii* this study | *Plotocnide borealis* | **9.4** | **8.5** | **0.7** | **0.4** |
| | *Margelopsis haeckelii* | 18.3 | 20.2 | 3.4 | 5.6 |
| | *Corymorpha nutans* | 20.2 | 21.3 | 3.1 | 5.1 |
| | *Euphysa aurata* | 18.6 | 17.9 | 3.1 | 5.2 |
| | *Hataia parva* | 19.9 | 18.9 | 2.1 | 4.6 |
| | *Protohydra leuckartii* | 16.6 | 18.9 | 1.6 | – |

*Euphysa brevia* (Uchida, 1947), *Euphysa flammea* (Hartlaub, 1902), *Euphysa japonica* (Maas, 1909), *Euphysa problematica* Schuchert, 1996), and some species of *Ectopleura*, such as *E. dumortierii* (Van Beneden, 1844) and *E. wrighti* Petersen, 1979. Genus *Euphysilla* with radially symmetrical medusae was recently transferred from Corymorphidae to Sphaerocorynidae (*Maggioni et al., 2021*). The *M. haeckelii* jellyfish has 3–4 tentacles per bulb instead of only one long tentacle or one long tentacle and three reduced per medusa, something typically seen among *Corymorpha* medusae (Table S2). Even in *Euphysa*, the sister group to *Corymorpha*, radially symmetric adult medusae may develop asymmetrically in contrast to medusae of *M. haeckelii*. The medusae of *E. flammea* only have a single tentacle in their youngest stage, with a second, third and fourth being added successively over time (*Schuchert, 2010*). Radially symmetric medusae in the species *P. borealis*, which is deeply nested in phylogenetic analyses of Aplanulata (*Pyataeva et al., 2016*; this study), suggests that radial symmetry has re-evolved in *M. haeckelii*, a manifestation of the original body plan symmetry for medusae of Aplanulata (Fig. 4).

According to molecular phylogenetic data, *M. haeckelii* was found within the corymphid clade together with *Corymorpha* and *Euphysa*. The taxonomic boundaries between genera *Corymorpha* and *Euphysa* are unclear. The genus *Corymorpha* is usually characterized by large, complexly arranged polyps, while the polyps of *Euphysa* are simply arranged (*Rees, 1957*). However, such a conclusion is premature, because hydroids are unknown for more than half of the 51 valid species of genus *Corymorpha*, and hydroids have been described for only three out of 13 valid species of genus *Euphysa*: *E. aurata*, *E. peregrina* and *Euphysa ruthae* Norenburg & Morse, 1983 (Table S1). For some jellyfish of the genus *Euphysa*, a *Corymorpha*-like hydroid is assumed (*Schuchert, 2010*) and *Corymorpha intermedia* Schuchert, 1996 nested within *Euphysa* clade and currently accepted as *E.*

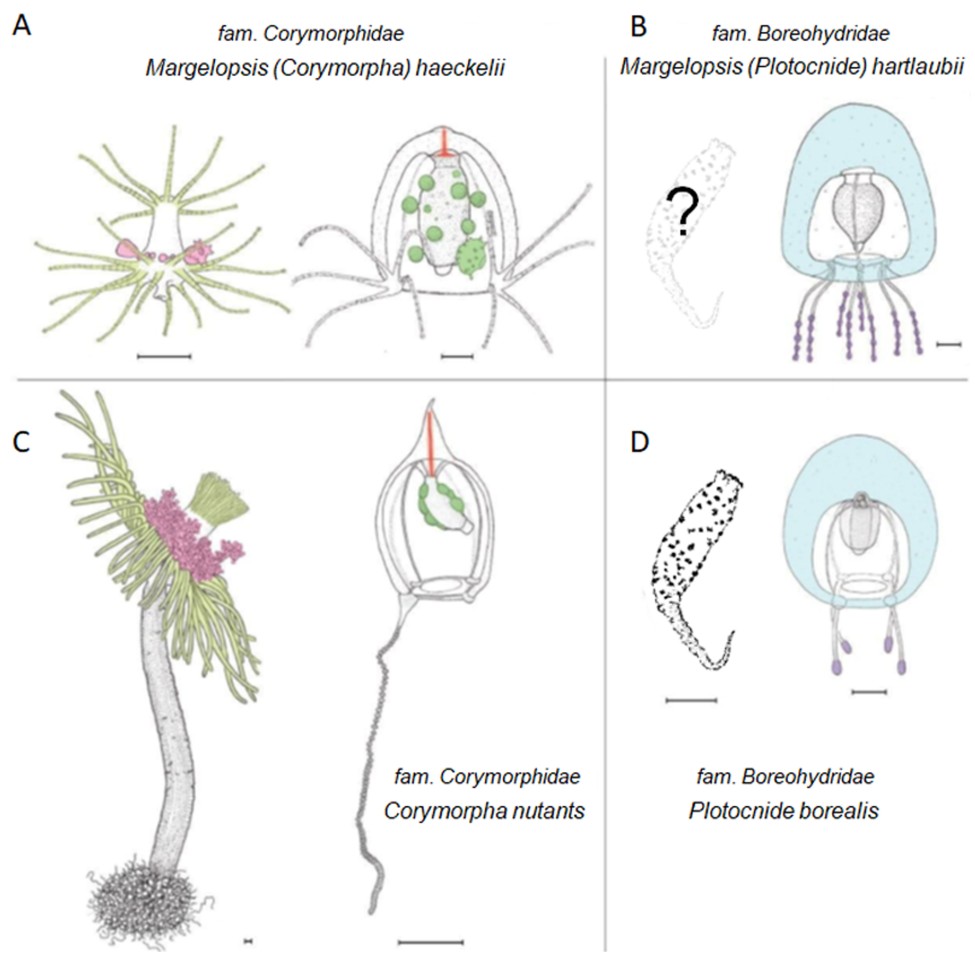

**Figure 3** Comparison of morphological characters of (A) *Margelopsis hartlaubii*, (B) *Margelopsis haeckelii*, (C) *Corymorpha nutans* and (D) *Plotocnide borealis*. Color coding: yellow, oral and aboral whorls of polyp tentacles; pink, region of medusa budding; green, the region of gametes formation; orange, apical canal; blue, medusa umbrella with clusters of exumbrellar nematoblasts; violet, clusters of nematocysts located at the distal parts of tentacles. *Margelopsis hartlaubii, Margelopsis haeckelii, Corymorpha nutans* and *Plotocnide borealis* modified from *Schuchert (2006)* and *Schuchert (2010)*. Scalebar –0.4 mm.

intermedia (*Nawrocki et al., 2013*). Nevertheless, the genera are clearly delimited by molecular phylogenetic analysis (*Nawrocki et al., 2013*; our data).

Although *M. haeckelii* forms a clade together with *Corymorpha* spp, its phylogenetic relationship with the species of the genus is not entirely clear. *Margelopsis haeckelii* may be related to *C. bigelow*, which forms a clade with *C. forbesii* on the phylogenetic tree 16S ($p = 1$; ML $=97$) (Fig. S2). But the relationship of the species *C. bigelow* and *C. forbesii* with *Corymorpha* spp. remains uncertain too. The differences of these species from the other *Corymorpha* spp. have already been noted on the basis of purely morphological data (*Petersen, 1990*). *Corymorpha bigelowi* differs in oral tentacles of polyps, and may be somewhat thickened at their tips (Table S1). Hydroids of *C. forbesii* also have moniliform oral tentacles. *Petersen (1990)* notes these two species (*C. bigelowi* and *C. forbesii*) have

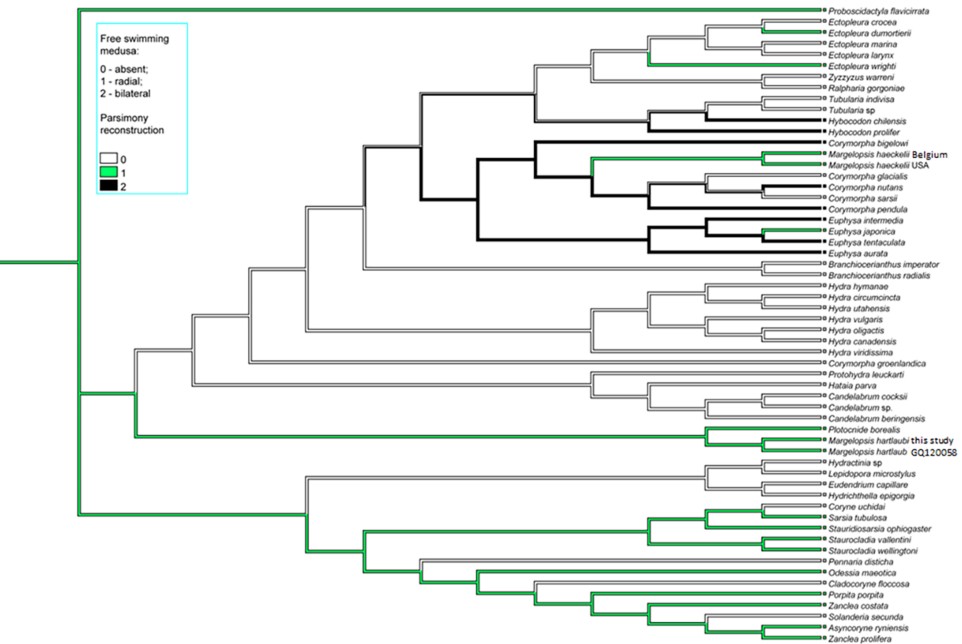

**Figure 4** **Parsimony ancestral state character reconstruction for medusae symmetry in Aplanulata.** Bayes Interference consensus tree for multigene dataset was used. Taxa were coded for no free-swimming medusa, bilateral symmetric medusa and radially symmetric medusa.

intermediate state of several characters such as arrangement of adhesive papillae and canals in hydrocaulus. In hydroids of genus *Euphysa* the hydrocauline papillae are located at the border between hydranth and hydrocaulus, hydroids lack canals in hydrocaulus. In *Gymnogonos* species only the aboral one-third of the hydrocaulus is provided with parenchyme and canals. Papillae are located in several close-set whorls just under the hydranth, scattered on the main part of the hydrocaulus and connected with the endodermal canals on the swollen basal part of the hydrocaulus. In *Corymorpha bigelowi* and *Corymorpha forbesii* only a limited number of simple endodermal canals run the length of the hydrocaulus and a limited number of papillae are lengthened basally to rooting filaments. The papillae are located on the swollen aboral part of the hydrocaulus. In typical *Corymorpha* species (such as *C. nutans*) the hydrocauline cavity is filled by parenchymatic endoderm with numerous peripheral longitudinal canals. Papillae and rooting filaments are located around the aboral end of the hydrocaulus. Unfortunately, *M. haeckelii* does not have a hydrocaulus and attachment papillae that would allow clarification of its phylogenetic position. *Margelopsis haeckelii* differs in that all tentacles of the hydroid are more or less capitate, but have ring-shaped or spiral batteries of nematocysts throughout their whole length (thus moniliform) as well hydroid has not hydrocaulus (*Werner, 1955*; *Schuchert, 2006*; Tables S1 and S2). The medusa of *M. haeckelii* is with four clusters of moniliform tentacles, without apical projection but with apical canal. Thus *M. haeckelii* has some characters similar with *C. bigelowi* and *C. forbesii* (nematocysts armament of tentacles in hydroid) and with *Corymorpha forbesii* (absence of apical process in medusa)

and some characters dissimilar with *Corymorpha* spp. (absence of hydrocaulus in hydroid, presence of clusters of tentacles in medusa).

*Margelopsis haeckelii* is grouped with the *Corymorpha*-clade, which is sister to *Euphysa*-clade. A prerequisite for the appearance of planktonic polyps *M. haeckelii* may be asexual reproduction through caulus or stolon segregation known for some corymorphids (for example, *Brinckmann-Voss, 1967*: Fig. 4). Polyps can temporarily detach from the substrate and be transferred by the current to a new habitat, as for example it is known for *E. peregrina* (*Rees, 1946*). Further studies and molecular phylogenetic data for more species of *Corymorpha* are needed to clarify the phylogenetic position of *C. forbesii* and *C. bigelowi*, as well as to clarify the relationship of *M. haeckelii* with the main clade *Corymorpha*. The composition of the genera *Corymorpha* and *Euphysa* and delimitation boundaries should be revised after the appearance of new molecular phylogenetic data for the genera *Corymorpha*, *Euphysa* and *Gymnogonos*.

## Validity of the species within the genus Margelopsis and the family Margelopsidae

*Margelopsis haeckelii* is the type species of the genus *Margelopsis*. Inclusion of *M. haeckelii* in the clade together with *Corymorpha* spp. calls into question the validity of the genus *Margelopsis* and the family Margelopsidae. This assumption is supported by the fact that most of the species in the family are poorly studied. To confirm the validity of Margelopsidae, new data on two more representatives of the family are needed: *Climacocodon ikarii* and *Pelagohydra mirabilis*. The life cycle and morphology of planktonic polyps *C. ikarii* is similar to *M. haeckelii* (*Kubota, 1979*). Both species were previously grouped into the subfamily Margelopsinae (*Rees & Ralph, 1970*). The hydroid of *Pelagohydra* with its enormous float is a much more complex organism than either *Margelopsis* or *Climacocodon*, being placed into the subfamily Pelagohydrinae (*Pilgrim, 1967*; *Rees & Ralph, 1970*). Perhaps the planktonic polyps of Pelagohydra originated independently of Margelopsinae and related to large corymorphid polyps with diaphragm of gastral cavity.

According to *WoRMS Editorial Board (2023)* there are three species within the genus *Margelopsis*: *M. haeckelii*, *M. hartlaubii* and *M. australis* (*Browne, 1910*). Validity of the fourth species, *M. gibbesii* (*McCrady, 1859*), is under consideration too (*Calder & Johnson, 2015*). Surprisingly, our concatenated gene dataset, as well as our each single gene (COI, 16S, 18S, 28S) dataset, recovered the medusa known as *M. hartlaubii* to be a close relative of *Plotocnide borealis,* and not closely related to *M. haeckelii* nor group within Corymorphidae. This result is further supported by independent morphological data showing several similarities between medusae of *M. hartlaubii* and *P. borealis,* including thick apical mesoglea of the bell (Fig. 3, marked blue), lack of an umbrella apical canal, nematocyst batteries being located at the distal parts of tentacles (Fig. 3, marked violet) and conspicuous nematocyst patches on the exumbrella in both species (*Schuchert, 2006*). Based on our findings, medusae described by *Browne (1903)* have been wrongly attributed to the genus *Margelopsis*. *Nawrocki et al. (2013)* suggested that the hypothesis of *M. hartlaubii* as the sister to the rest of Aplanulata was uncertain due to low bootstrap support and

that more genetic markers were needed to understand the phylogenetic placement of the species. Based on our multi-marker phylogenetic analysis and morphological data (*Browne, 1903*; *Schuchert, 2006*) we hypothesize that *M. hartlaubii* has a mud-dwelling, meiobenthic polyp like *P. borealis* (Fig. 3), and that the two species combined represent the sister group to the rest of Aplanulata. Given the similarity in morphology of medusae of *M. hartlaubii* and *P. borealis* and relatively small *P-distances* (Table 2), the species *M. hartlaubii* should be moved to the genus *Plotocnide* after the similarity of their polyps will be confirmed.

In addition to *M. haeckelii* and *M. hartlaubii,* there are several other poor studied species in the genus *Margelopsis,* including *Margelopsis gibbesii* and *Margelopsis australis.* Following *Schuchert (2006)*, the World Register of Marine Species (*WoRMS Editorial Board, 2022*) lists *Margelopsis gibbesii* as invalid. This stems from the fact that the original material used to describe this species as *Nemopsis gibbesii*, consisted of a margelopsid polyp and a bougainvilliid medusa, the latter subsequently recognized as a medusa of *Nemopsis bachei* (L. Agassiz, 1862). This situation has generated subsequent nomenclatural confusion. More recently, *Calder & Johnson (2015)* stabilized the situation by designating the hydroid specimen illustrated by *McCrady (1859)* in Plate 10, Figure 7 as a lectotype for the margelopsid species. *Calder & Johnson (2015)* went on to provide evidence casting doubt on the distinction between *M. gibbesii* and *M. haeckelii* but maintained the two species given the geographic locations on either side of the north Atlantic and pending further study. In the species *M. hartlaubii*, male medusae have been described, unlike *M. haeckelii*, for which parthenogenetic development is assumed (*Calder & Johnson, 2015*). Other distinguishing features for medusae of the two species are the number of the tentacles on the marginal bulbs, the width of the apical canal, and color of the manubrium and marginal bulbs. On the other hand, there are doubts about the validity of these differences (*Schuchert, 2006*; *Calder & Johnson, 2015*). In this study, however, using molecular phylogenetics, we have shown that medusae of *M. haeckelii* can be collected at both sides of the North Atlantic. The lack of any nucleotide substitution in COI and 16S sequences of *M. haeckelii* representatives from both sides of Atlantic Ocean makes it possible to suggest that these two populations are not isolated. This finding supports the hypothesis that species *M. gibbesii* is invalid. However, to confirm the hypothesis, it is necessary to obtain a sequence of a sample that will be reliably identified as *M. gibbesii* according to morphological data, for example, a male medusa caught off the North Atlantic coast of America.

*Margelopsis australis* is only known from its original collection and is based on a single medusa specimen, lacking reliable characters for distinguishing it from *M. hartlaubii* (*Browne, 1910*). Moreover, the single specimen *M. australis* was described as being "somewhat contracted and in a crumbled condition" (*Browne, 1910*). Arrangement of the nematocysts upon the tentacles of *M. australis* is unknown because tentacles of studied specimen were closely contracted. The only difference is, unlike *M. hartlaubii*, the exumbrella of *M. australis* is covered with isolated nematocysts which are not arranged in groups. Based on the available morphological data, we cannot state with any degree of certainty that *M. australis* is a valid species, or that it is a member of Boreohydridae or Corymorphidae.
## CONCLUSION

Our results clarify the phylogenetic picture of Aplanulata by revealing the phylogenetic position of *M. haeckelii,* type species of the genus *Margelopsis* as falling within *Corymorpha* and *M. hartlaubii* as being a close relative of *Plotocnide* in the family Boreohydridae. In the case of the latter species, this phylogenetic result conflicts with the century-old hypothesis that *Margelopsis* belongs to Tubulariidae or Corymorphidae (*Nawrocki et al., 2013*). However, by showing that *M. haeckelii* falls within the genus *Corymorpha*, our investigation presents strong evidence in support of this traditional hypothesis. Because *M. haeckelii* is a hydrozoan belonging to Corymorphidae, we can infer that this lineage evolutionarily lost their hydrocaulus and stolon, likely as an adaptation to a holopelagic life cycle. It was previously suggested that the foundation for this type of change in body plan, and accompanying lifestyle, might lead to speciation and could be reflected by changes in the expression of Wnt signaling components (*Duffy, 2011*). Based on our results, *M. haeckelii* might be a prime candidate for testing this hypothesis.

Unfortunately, due to the few and extremely irregular documented collection records of hydroids from the supposedly sister genera of *Margelopsis*, *Pelagohydra* and *Climacocodon*, the phylogenetic relationships within this group are still obscured. It remains unclear if *Pelagohydra* and *Climacodon* form a clade with either *M. hartlaubii* or *M. haeckelii*, or neither. Portions of this text were previously published as part of a preprint (*Kupaeva et al., 2022*).

## ACKNOWLEDGEMENTS

We thank Dr. Peter Schuchert for the gift of *Margelopsis hartlaubii* DNA and *Margelopsis hartlaubii* medusa sample image. We are grateful to Dr. Allen G. Collins for sharing with us *Margelopsis haeckelii* (USA) sequences. We also thank Dr. Brett Gonzales for the help with text and grammar editing. We like to acknowledge the staff of N.A. Pertzov White Sea Biological Station of Lomonosov Moscow State University, Russia, for providing opportunity for the research and equipment usage of the Center of microscopy WSBS.

### Funding

This study was supported by federal project 0088-2021-0009 of the Koltzov Institute of Developmental Biology of the RAS (Stanislav Kremnyov) and the Scientific Project of the State Order of the Government of Russian Federation to Lomonosov Moscow State University, grants No. 121032300118–0 (Andrey Prudkovsky) and No. 121032300066-4 (Stanislav Kremnyov). The funders had no role in study design, data collection and analysis, decision to publish, or preparation of the manuscript.

### Grant Disclosures

The following grant information was disclosed by the authors:

federal project 0088-2021-0009 of the Koltzov Institute of Developmental Biology of the RAS.

Scientific Project of the State Order of the Government of Russian Federation to Lomonosov Moscow State University: 121032300118-0, 121032300066-4.

## Competing Interests

The authors declare there are no competing interests.

## Author Contributions

- Daria Kupaeva conceived and designed the experiments, performed the experiments, analyzed the data, prepared figures and/or tables, authored or reviewed drafts of the article, and approved the final draft.
- Tatiana Lebedeva performed the experiments, authored or reviewed drafts of the article, and approved the final draft.
- Zachariah Kobrinsky performed the experiments, authored or reviewed drafts of the article, and approved the final draft.
- Daniel Vanwalleghem performed the experiments, authored or reviewed drafts of the article, and approved the final draft.
- Andrey Prudkovsky conceived and designed the experiments, performed the experiments, analyzed the data, prepared figures and/or tables, authored or reviewed drafts of the article, and approved the final draft.
- Stanislav Kremnyov conceived and designed the experiments, performed the experiments, analyzed the data, prepared figures and/or tables, authored or reviewed drafts of the article, and approved the final draft.

## Data Availability

The sequences obtained in this study are available in GenBank: OK129327, OK139084, OK142735, OK127861, ON391039, ON391070, ON237369, ON237671, ON237710, OK110252.

## Supplemental Information

Supplemental information for this article can be found online at http://dx.doi.org/10.7717/peerj.16265#supplemental-information.

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
