# Peer review of "Margelopsid species search taxonomic home within Corymorphidae and Boreohydridae"

_PeerJ, doi:10.7717/peerj.16265_

## Round 0.1 · original submission · Major Revisions

Dear authors,

As you can see, there are three reviews for your manuscript, two of them recommending minor revision, and one with a request to reject the manuscript. Therefore, I draw your special attention to the remarks of this third reviewer. Please carefully respond to all their, as well as the others' comments.

·

Basic reporting

The language used throughout the manuscript is overall clear. However some rephrasing are necessery for the comprehension of the manuscript.
The literature used in this manuscript is clearly referenced but requires additional sources to support part of the manuscript that appear speculative in the current state.
The manuscript well structured in accordance to the standards of a scientific publication. Figures and tables are clear but some figures and their associated legends require some corrections. The raw data and/or source of the data are shared.
This manuscript is self-contained and does not overstate its results, however some of the interpretations of results require further clarification on the data.

Experimental design

The manuscript is within the aims and scope of the journal.
The research question is well defined and properly investigated with rigorous tools and approaches.
Some aspects of the methods need to be devloped in order for these results to be replicated.

Validity of the findings

The rationale and benefit to the literature is clearly stated.
The data presented in this manuscript are robust according to the standards in the field.
The conclusions are well stated, clearly answer the research question and do not overstate the findings of the manuscript.

Reviewer 2 ·

Basic reporting

I am conflicted about this paper. Although the findings are interesting, the paper is lacking in some crucial aspects.

Experimental design

The authors sequenced two/three individuals belonging to two Margelopsis species and show that they fall into two different clades; one falls within Corymorpha, and the other is sister to Plotocnide borealis. The take-home message of the paper is that Margelopsis does not seem to be a monophyletic genus, and the two Margelopsis species possibly belong to two different families.
In my opinion, to draw any conclusion, you need more representative of the species of interest and the genera and families as well. There are a lot of genera not represented in this tree, and although the results are interesting, I would strongly advise against proposing new nomenclature. Too many missing species and too few individuals for each species of interest impedes a solid conclusion on which to base new taxonomic proposals and recommendations. For example, the family Corymorphidae contains ten genera, but only three are represented in this tree. Only two Margelopsis haekelii individuals were sequenced. This is not enough. Also, the family Margelopsidae contains three genera, two of which are not present in this tree.
Likewise, only two specimens of M. hartlaubii are available. The family they seem to fit in (Boreohydridae) contains two genera (3 species total), and only one is available in this tree.

Validity of the findings

The paper is oddly missing some crucial information but has a lot of irrelevant information.
In "material and methods", it is not clear how many individuals were collected, how many were sequenced, and whether collecting permits were necessary. How big was the final dataset? How many sequences? Did the authors download from Genbank all the relevant sequences?
The discussion needs to be cut substantially. It contains information marginal to the actual findings and it requires significant restructuring. For example, the first paragraph focuses on the species Euphysa peregrina and Eirene, which are not even present in the phylogenetic tree. Reading the discussion, I was lost trying to figure out the relevance of most of the information.
Bottom line, in my opinion, the discussion should be reduced by a lot, leaving only relevant information, presented in a way that highlights the paper's findings.

Additional comments

In summary, the findings of the paper are, as it stands, limited. With one or two new individuals of Margelopsis/species sequenced, the authors show that Margelopsis may not be a monophyletic genus and that possibly M. hartlaubii falls with Plotocnide, and M. haeckelii within a Corymopha clade. Based on the little new data presented and the general quality of the paper, I suggest the paper is rejected. The authors should try to refocus and sharpen the paper, describe the methods better, rescale the results, cut on unnecessary discussion, and refrain from proposing new taxonomy. After these changes, the paper could be resubmitted to appropriate journals.

Reviewer 3 ·

Basic reporting

The article "Margelopsid species search taxonomic home within
Corymorphidae and Boreohydridae" contribute significantly to elucidating members of the Hydrozoa. In addition, it makes use of integrative taxonomy. It is clear, and the author used good literature.

Experimental design

Specifying some aspects of the specimens used and analyses performed is necessary.

Validity of the findings

The article contribute significantly to elucidating members of the Hydrozoa

Additional comments

The authors will find specific comments in the PDF that must be addressed.
Although they include some aspects of nematocysts, it is suggested to complement the results with cnidome analysis of the organisms involved or to mention the relevance of knowing it for future work.

Annotated reviews are not available for download in order to protect the identity of reviewers who chose to remain anonymous.

---

## Round 0.2 · accepted · Accept

Dear Dr. Kremnyov,

I have assessed the revision myself, and I am happy with the current version. Thank you for the detailed responses to the comments of the reviewers and for the corrections made to the manuscript. Despite the harsh criticism from Reviewer #2, I found your answers to be quite correct and consistent with what the reviewer wanted from you.

Now your manuscript is ready for publication.

Best regards,
Alexander